# EEG Evidence of Acute Stress Enhancing Inhibition Control by Increasing Attention

**DOI:** 10.3390/brainsci14101013

**Published:** 2024-10-10

**Authors:** Bingxin Yan, Yifan Wang, Yuxuan Yang, Di Wu, Kewei Sun, Wei Xiao

**Affiliations:** Department of Military Medical Psychology, Air Force Medical University, Xi’an 710032, China; 15927177279@163.com (B.Y.); wyyff_1220@163.com (Y.W.); 13903442056@163.com (Y.Y.); 18702980966@163.com (D.W.); xlxsunkewei@126.com (K.S.)

**Keywords:** acute stress, inhibitory control, Stroop, EEG, frequency domain analysis, microstate analysis

## Abstract

Objective: Research about the impact of acute stress on inhibitory control remains a contentious topic, with no consensus reached thus far. This study aims to investigate the effects of acute stress on an individual’s inhibitory control abilities and to elucidate the underlying neural mechanisms by analyzing resting state electroencephalogram (EEG) data. Methods: We recruited 32 male college students through participant recruitment information to undergo within-subject experiments under stress and non-stress conditions. Physiological indicators (cortisol and heart rate), self-report questionnaires, and behavioral data from the Stroop task were collected before, during, and after the experiment. Additionally, a five-minute eyes closed resting state EEG data collection was conducted during the Stroop task before. Results: (1) Acute stress led to a reduction in the conflict effect during the participants’ Stroop task in individuals. (2) Stress resulted in an increase in the power of the beta in the resting state EEG. (3) Acute stress caused an increase in the duration of class D and an increase in the transition probabilities from classes C and B to class D in the microstates of the resting state EEG. (4) Acute stress leads to an increase in beta power values in individuals’ resting state EEGs, which is significantly negatively correlated with the reduction of the conflict effect in the Stroop task under stress. Conclusions: Acute stress can enhance individuals’ attentional level, thereby promoting inhibitory control performance.

## 1. Introduction

Acute stress is a common occurrence in life, often triggered by situations such as interviews and exams, leading to feelings of tension in individuals. Stress refers to a series of physiological and psychological responses that an individual undergoes to maintain homeostasis when faced with external threats [1]. From a physiological standpoint, individuals confront stress by engaging in a series of physiological responses to maintain homeostasis and better cope with stressful situations. Physiologically, individuals facing stress undergo a series of physiological responses to maintain homeostasis and more effectively manage the stressor. These responses are primarily due to the activation of the sympathetic nervous system (SNS), which leads to increased heart rate and blood pressure, and the activation of the hypothalamic–pituitary–adrenal (HPA) axis, which results in the elevation of hormones such as cortisol [2]. The SNS axis is often referred to as the “fast axis”, because it quickly induces the “fight or flight” response, with the increases in heart rate and blood pressure typically returning to normal within minutes. In contrast, the HPA axis is commonly known as the “slow axis”, as stress triggers the release of glucocorticoids (such as cortisol) from the HPA axis, which usually peak 15–25 min after stress induction and return to baseline around an hour later [3]. Stress has been demonstrated to affect various higher cognitive activities in individuals, including decision-making and working memory, as well as the brain regions associated with these cognitive functions [4,5,6]. Therefore, stress is closely related to an individual’s cognitive performance, but the underlying neural mechanisms remain not fully understood.

Inhibition control is an important component of an individual’s executive functions [7]. As is defined, it is the ability to suppress irrelevant, distracting, incorrect, or inappropriate dominant responses or thoughts, impulses, cognitive conflicts, behavioral choices, and automated behavioral habits [8], which is crucial for the normal functioning of higher cognitive activities [9]. Better inhibitory control helps individuals resist negative temptations, inhibit inappropriate behaviors, and make more appropriate decisions. Conversely, a lack of inhibitory control can lead to adverse effects: alcohol addiction, gaming addiction, obesity, and even neurological disorders such as depression and attention deficit hyperactivity disorder (ADHD) are linked to insufficient inhibitory control [10,11,12]. The Stroop task is a classic paradigm for measuring inhibitory control [13], which involves presenting stimuli that are either congruent or incongruent in terms of color and meaning, and requires subjects to make judgments based on the font color or meaning presented [14]. Due to the complex conflicts between stimuli in the Stroop task, this study chooses the Stroop task as a standard for measuring individual inhibitory control.

Current research suggests that acute stress may affect inhibitory control, but research on the impact of acute stress on inhibitory control is still contentious: most past studies believe that, during acute stress, individuals need to mobilize cognitive resources to cope with emergencies, which can impair their inhibitory control [15,16], while some studies argue that appropriate stress can enhance an individual’s inhibitory control [13,15,16,17,18]. There is no definitive conclusion on whether acute stress promotes or impairs inhibitory control behavior, and there is a need for further research on the mechanisms by which acute stress affects an individual’s behavioral performance. Therefore, the purpose of this study is to understand whether acute stress can promote or impair individuals‘ inhibitory control.

Electroencephalography (EEG) is considered a tool for exploring the electrophysiology of the brain, characterized by its extremely high temporal resolution. EEG frequency domain analysis can be used to analyze the oscillatory activity of the cerebral cortex, studying the relationship between brain activity and individual state characteristics based on the characteristics of distinct oscillations [19,20,21]. It is believed that the brain’s oscillatory activity changes in different spatiotemporal patterns during different cognitive activities, mainly divided into delta, theta, alpha, beta, and gamma oscillations based on frequency [20,22]. There is a lack of research on the relationship between stress and different frequency EEG oscillations, so we aim to provide a new perspective on the neural mechanisms of stress’s impact on inhibitory control by examining the changes in different frequency EEG oscillations before and after stress and the relationship between these changes and the changes in behavioral performance.

In recent years, numerous studies have utilized functional magnetic resonance imaging (fMRI) to explore brain regions and network changes associated with stress. However, fMRI is more complex and costly compared to EEG, prompting our attempt to investigate stress-related brain network activity using EEG technology. Microstate analysis is an excellent research method for this purpose. Microstate analysis is an EEG technique that examines brain changes at the millisecond level by presenting the topographical distribution of EEG signals. It assesses multi-channel EEG recordings as a series of quasi-stable microstates, making it suitable for evaluating sub-second changes in brain activity [23]. Consequently, some researchers consider EEG microstates to be an ideal method for studying the temporal dynamics of large-scale brain networks following acute stress [24]. This analysis involves segmenting EEG activity into periods of approximately 50–120 ms, during which, synchronized neural networks generate consistent spatial potential topographies on the scalp [25]. Currently, it is common to categorize the spatial potential topographies generated by microstates into four types: A, B, C, and D. These categories are observable in EEGs and account for 65–84% of the global variance of the data [26,27]. It is widely believed that these four categories of microstates are related to certain brain networks [28]: Class A is associated with activations in the bilateral superior and middle temporal gyri, which are involved in phonological processing, speech, and auditory processing [29]. Class B is associated with bilateral extravasate visual areas that have been identified as the visual networks [30]. Class C is associated with activations in the dorsal anterior cingulate cortex, the bilateral inferior frontal cortices, and the insula, which are related to the salient network [31]. Class D is associated with signaling in the right-lateralized dorsal and ventral areas of the frontal and parietal cortices, which are related to ventral front–parietal attentional networks and are associated with the switching and reorientation of attention [32]. Therefore, we aim to utilize the four categories of microstate analysis to better elucidate the brain networks associated with individual EEG activity under stress, thereby further revealing the underlying mechanisms.

In summary, this study primarily aims to investigate the impact of acute stress on individual inhibitory control levels, providing evidence for previous debates on the relationship between acute stress and inhibitory control. Additionally, this study seeks to explore the neural network mechanisms related to acute stress’s impact on inhibitory control, discussing the neural mechanisms behind these behavioral changes and thereby providing more possibilities for improving individual inhibitory control performance under stress.

## 2. Materials and Methods

### 2.1. Participants

Through campus recruitment channels, we posted participant recruitment information and, after a selection process, enrolled 34 male college students from both our institution and external schools, including 6 students from our university and 28 from other universities. Participant selection criteria were as follows: male (female participants were not included due to the potential impact of hormonal fluctuations on cortisol concentration levels), aged between 18 and 30, right-handed, in good health without significant diseases such as heart conditions, free from neurological or psychiatric disorders, without oral diseases such as mouth ulcers, having not participated in any psychology-related experiments before, without color vision deficiencies, with a BMI within the normal range, and with normal or corrected-to-normal vision. Exclusion criteria included a score not exceeding 28 on the Perceived Stress Scale (PSS-10) and a score not exceeding 50 on the Self-Rating Anxiety Scale (SAS) (to ensure participants were not under high stress or experiencing anxiety recently) and withdrawal from the experiment midway through any of the two sessions. After the experiment, it was discovered that one participant’s resting state EEG data under stress conditions were missing, and during data analysis, another participant was found to have excessive noise in their resting state EEG, leading to the exclusion of both participants’ data. Ultimately, data from 32 participants were included in the analysis. This study was approved by the Ethics Committee of Xijing Hospital (2024–06–03), was registered at Clinical Trials.gov (KY20242146-F-1), and was performed following the ethical standards as laid down in the 1964 Declaration of Helsinki and its later amendments or comparable ethical standards, and all participants voluntarily took part in the experiment after signing an informed consent form.

### 2.2. Materials

#### 2.2.1. Task

The study measures individuals’ inhibitory control levels by the Stroop task, which is one of the commonly used tasks for assessing interference suppression. The task program was developed using E-Prime 3.0 and presented on a 37 × 30 cm computer screen. The stimuli in the task can be categorized into four types based on word meaning: “red”, “blue”, “green”, and “yellow”, with each word meaning matched with one of the four different colors and presented randomly (the ratio of color–word consistent trials to inconsistent trials is 1:3). There are a total of 16 task stimuli, with each consistent stimulus presented 30 times and each inconsistent stimulus presented 10 times, totaling 120 trials for each type. Participants are required to select the corresponding key (“f”, “g”, “j”, or “k”) based on the actual color of the presented word (“red”, “blue”, “green”, or “yellow”) and respond by pressing the key. The reaction times for both consistent and inconsistent trials are recorded, and the conflict effect and conflict score are calculated; both are essential indicators of participants’ inhibitory control performance in the Stroop task. The conflict effect = reaction time for inconsistent trials − reaction time for consistent trials, and the conflict score = conflict effect/reaction time for consistent trials. The smaller the conflict effect and conflict score, the less susceptible the participants are to interference from word meaning when selecting based on color, indicating more vital inhibitory control ability to manage interference [33].

At the beginning of the experiment, a comprehensive set of instructions appears on the screen. Participants press any key to start training after ensuring they understand the task rules. The practice consists of 5 trials with no time limit on stimulus presentation, and feedback is given on whether the critical press is correct. After the practice, participants press the spacebar to enter the formal experiment. In the formal experiment, a fixation point is presented in the center for 1000 ms before the stimulus image is presented in the center of the screen for 800 ms, followed by a 1000 ms blank screen. Participants are required to respond by pressing a key before the blank screen disappears, and no feedback is given on the correctness of the response. Due to the lengthy duration of the complete experiment, the formal experiment is divided into two equal parts, with 240 stimuli randomly divided into two parts, each presenting 120 trials (referred to as Stroop1 and Stroop2 in the following text). After the first part, participants can take a short break (during the stress condition, participants need to collect saliva during this time) before proceeding to the second part. The experiment ends only after both parts are completed. The specific procedure of the Stroop task is illustrated in Figure 1.

#### 2.2.2. Acute Stress-Evoking

The typical laboratory stress induction paradigms include physiological, cognitive, and integrated stress induction paradigms [34]. To ensure stress intensity, most studies now opt for integrated stress induction paradigms that combine cognitive and psychological elements to induce acute stress, such as the Trier Social Stress Test (TSST) and the Socially Evaluated Cold Pressor Test (SECPT).

This study employs the SECPT paradigm [35] to induce acute stress in participants. Numerous studies have found that SECPT can elevate participants’ subjective stress levels, heart rate, and cortisol concentrations, indicating that SECPT is an effective paradigm for inducing acute stress in the laboratory [36,37]. During the stress process, participants are asked to submerge their right hand completely in ice water at 0–4 °C for three minutes. A camera is placed in front of participants, and they are informed that the entire process will be videotaped for subsequent facial expression analysis experiments. Concurrently, a male and a female researcher sit before the participant to observe and record the entire process.

Indicators of stress induction intensity include heart rate, cortisol concentration, subjective stress levels, and scores on the positive and negative affect scales.

Heart rate is recorded throughout the experiment using a POLAR pacer wristband (Polar (Guangzhou) Electronics Co., Ltd., Guangzhou, China) heart rate monitor. Previous studies have shown that cortisol concentration peaks 20–30 min after acute stress [38,39]. Therefore, we collect saliva samples from participants before stress and at 10, 20, 30, and 40 min after stress induction using cortisol collection tubes. Participants are asked to keep a cotton swab from the collection tube in their mouth for 2 to 3 min and then place it in a −20 °C freezer for later extraction and cortisol concentration measurement using a salivary cortisol ELISA kit at Xi’an Haoyang Biotechnology Company (Xi’an, China); subjective stress levels are assessed by asking participants if they feel stressed and rating the perceived stress on a scale from 0 to 10 after the experiment.

#### 2.2.3. Related Scales

Scale of Positive and Negative Experience (SPANE): This scale consists of 12 items, primarily used to measure individuals’ subjective emotional experiences, with 6 items measuring positive experiences and 6 items measuring negative experiences. The scale uses a 5-point scoring system, with scores for positive and negative emotional experiences ranging from 6 to 30 points each. Additionally, an emotional experience balance score can be obtained, ranging from −24 to 24 points [40].

### 2.3. Experimental Procedure

The experiment is designed as a within-subjects study. To avoid practice effects, each participant must undergo one acute stress and one non-stress experiment on two separate days, with at least one week between them, and the order of stress and non-stress experiments is entirely randomized. The results obtained from the non-stress experiment are considered the baseline performance level of the participants. After the recruitment and screening of participants, the principal investigator must remind them of the experiment time one day in advance and ask them to avoid vigorous exercise and eating two hours before the experiment. They are also asked to wash their hair before attending the experiment to keep the scalp clean, avoid wearing metal head ornaments or earrings, and maintain oral hygiene. To avoid the influence of cortisol’s circadian rhythm, all experiments are conducted between 2 pm and 6 pm, and participants are ensured to have a sleep duration of at least 7 h the night before.

Acute Stress Experiment: Upon arrival at the laboratory, participants wear a heart rate belt, wristwatch, and other equipment; sit quietly for five minutes; and complete the SPANE for the first time. Afterward, the first saliva sample is collected (T0) and heart rate recording begins. Following this, an EEG cap is fitted to the participant, and after completion, the participant undergoes acute stress induction. After the induction, the participant fills out the SPANE for the second time, then follows a five-minute eyes closed resting state EEG data collection. After this, the second saliva sample is collected (T1). Subsequently, the participant completes the first part of the Stroop task (approximately 8 min long), and after completion, the third saliva sample is collected (T2). After the collection, the participant proceeds with the second part of the Stroop task (approximately 8 min long), and after completion, the fourth saliva sample is collected (T3). After the tasks, the participant sits quietly for ten minutes before the fifth saliva sample is collected (T4) and completes the third SPANE. Afterward, participants are asked if they felt anxious during the SECPT phase and rate the level of nervousness on a scale from zero to ten (zero being not anxious at all, and ten being unbearably nervous).

Non-Stress Experiment: Upon arrival at the laboratory, participants sit quietly for 5 min before an EEG cap is fitted. After the fitting, the participant undergoes a 5-min eyes closed resting state EEG data collection. After the collection, the participant completes both parts of the Stroop task and is then asked if they felt any physical discomfort or pain during the experiment. Figure 2 shows the entire experimental procedure.

### 2.4. Recording of EEG

In our study, electroencephalogram (EEG) data were collected using a 32-channel EEG device manufactured by Brain Products (Gilching, Germany), with the Mitsar-EEG-202 amplifier and WinEEG software 2.8 (St. Petersburg, Russia). The EEG data were collected with a default sampling rate of 1000 Hz, utilizing a 32-channel electrode cap based on the international 10–20 system standard. The online reference was set at FCz, with GND as the ground. Recording of resting state EEG signals commenced once the impedance between the reference electrode and the scalp was reduced to below 10 kΩ and the impedance between all other electrodes and the scalp was below 20 kΩ. Participants were instructed to keep their eyes closed and remain awake throughout the recording and were advised to minimize eye and head movements. The recording duration was approximately five minutes.

Offline data preprocessing was conducted using the MATLAB2013 plugin EEGLAB12.1.1: continuous EEG data were band pass-filtered between 0.1 and 40 Hz using a FIR filter, with a notch filter applied for frequencies between 48 and 52 Hz, maintaining a sampling rate of 1000 Hz. The reference electrode for offline analysis was shifted to the bilateral mastoids TP9/TP10. The data were then segmented into 2000 ms lengths, after the manual removal of noise with a large floating amplitude and high frequency, Independent Component Analysis (ICA) was performed to eliminate ocular and blink artifacts. Finally, extreme values exceeding ±100 μV were removed, completing the preprocessing of the resting state EEG.

### 2.5. Frequency Domain Analysis

Utilizing the EEGLAB plugin, Fast Fourier Transform (FFT) was applied to the preprocessed resting state data to obtain the power values across all frequency bands and channels for stress and non-stress conditions. By extracting the data according to the delta (0–4 Hz), theta (4–8 Hz), alpha (8–12 Hz), and beta (12–30 Hz) bands, we derived the power values for each frequency band across the entire brain, and topographic maps were plotted to see whether stress might influence the power values of each frequency band.

### 2.6. Microstate Analysis

Following the four microstate categories commonly utilized in prior research [21,28], the data were subjected to clustering analysis. EEG data from all participants were analyzed using k-means clustering and iterative classification, resulting in the extraction and categorization of all momentary maps at the peaks of Global Field Power (GFP) into four distinct map topographies. Paired sample *t*-tests were employed to analyze the duration, occurrence, and contribution of the four microstates, as well as the transition probabilities between different microstate categories, under stress and non-stress conditions. This approach examined the differences in microstates and their interconversion across varying states.

### 2.7. Data Analysis

#### 2.7.1. Analysis of Stress-Related Indicators

A one-way ANOVA and post hoc tests were conducted on the baseline cortisol concentration, the peak cortisol concentration after stress induction, and the cortisol concentration at the end of the experiment. A one-way ANOVA and post hoc tests were also performed on the mean heart rate 3 min before stress, the mean heart rate 3 min after stress induction, and the heart rate at the end of the experiment. The scores for subjective stress levels were analyzed using a one-sample *t*-test. The total scores, positive and negative affect scores, and balance scores of the SPANE scale before stress, after stress induction, and after task completion were analyzed using a one-way ANOVA and post hoc tests to determine whether the SECPT paradigm successfully induced acute stress.

#### 2.7.2. Behavioral Analysis

Paired-sample *t*-tests were conducted to analyze the conflict effect and conflict score of the Stroop task under stress and non-stress conditions to determine if there was a significant difference in inhibitory control ability under different states.

#### 2.7.3. Frequency Domain and Microstate Analysis

A paired-sample *t*-test was conducted to analyze the power values of each frequency band in the frontal brain regions (Fp1, Fp2, F3, F4, F7, F8, Fz, Fc1, Fc2, Fc5, and Fc6) to determine if there were any differences in EEG power values under stress and non-stress conditions. In addition, paired-sample *t*-tests were also used to analyze the duration, occurrence, and contribution of the four microstates, as well as the transition probabilities between different microstate categories under stress and non-stress conditions, to examine the differences in microstates and their interconversion under different states.

#### 2.7.4. Regression Analysis

Regression analysis was performed on the differences in power values of the oscillations that showed significant differences under stress and non-stress conditions, and the differences in conflict effect and conflict score, to analyze whether the changes in oscillatory power values caused by stress are related to changes in stress-induced inhibitory control ability.

## 3. Results

### 3.1. Stress Induction

#### 3.1.1. Subjective Measurements

One-sample *t*-test of subjective stress levels revealed that participants generally felt stressed during the stress induction process (t = 12.988, *p* < 0.001). Regarding the SPANE scale scores, although there was no significant difference in the total SPANE scores before and after stress and at the end of the experiment (F(1,31) = 0.584, *p* > 0.05, *ηp*^2^ = 0.019), significant differences were observed in positive emotions (F(1,32) = 4.2, *p* < 0.01, *ηp*^2^ = 0.089), negative emotions (F(1,31) = 4.554, *p* < 0.01, *ηp*^2^ = 0.046), and balance scores (F(1,32) = 5.777, *p* < 0.01, *ηp*^2^ = 0.080) before and after stress. Significant differences were also found in positive emotions (t = 3.088, *p* < 0.05), negative emotions (t = −2.294, *p* < 0.05), and balance scores (t = 5.558, *p* < 0.01) between the pre-stress and post-stress periods, as well as in negative emotion scores (t = 1.323, *p* < 0.05) between post-stress and the end of the experiment. The SECPT paradigm led to a decrease in positive emotions and an increase in negative emotions among participants, who generally felt stressed. The SPANE scale results indicate that the SECPT significantly elevated negative emotions and reduced positive emotions in this study. The result of the SPANE score is shown in Figure 3.

#### 3.1.2. Salivary Cortisol

Saliva samples collected were analyzed by the Xi’an Haoyang Biotechnology Company to obtain cortisol concentrations at different time points. Due to individual differences in the timing of cortisol concentration peaks, a one-way ANOVA was conducted on the cortisol concentrations before stress (which we consider to be the baseline cortisol level, previously mentioned as T0), the maximum cortisol concentration at 20 and 30 min after stress (which we consider to be the peak cortisol concentrations), and the cortisol concentration at 40 min after stress (the cortisol concentration at the end of the experiment, previously mentioned as T4). The results showed a significant difference (F(1,31) = 13.10, *p* < 0.001, *ηp*^2^ = 0.22). Further post hoc testing using Dunnett’s test revealed that the peak cortisol concentration at 20–30 min after stress was significantly higher than the baseline level (t = 6.890, *p* < 0.001) and the cortisol concentration at the end of the experiment (t = 6.723, *p* < 0.001), indicating that stress successfully induced an increase in cortisol concentration.

#### 3.1.3. Heart Rate

The heart rate of the participants was recorded for three minutes before stress induction to obtain the baseline heart rate level. The heart rate was again averaged over three minutes at the end of the stress period to obtain the peak heart rate. Additionally, the heart rate was averaged over three minutes at the end of the experiment, 40 min after stress, to acquire the heart rate after the study. A one-way ANOVA was conducted on the heart rate values from these three time periods, and the difference was found to be significant (F(1,31) = 13.75, *p* < 0.001, *ηp*^2^ = 0.217). Further post hoc testing using the LSD test revealed that the heart rate after stress was significantly higher than the baseline heart rate (t = 12.232, *p* < 0.001) and the heart rate at the end of the experiment (t = 12.052, *p* < 0.001), indicating that stress successfully induced a significant increase in heart rate. The result of salivary cortisol and heart rate is shown in Figure 4.

### 3.2. Behavioral Results

A paired-sample *t*-test was conducted to compare the conflict effect and conflict score of the Stroop task under stress and non-stress conditions. The results revealed significant differences in the conflict effect (t = −2.80, *p* < 0.01) and conflict score (t = −2.35, *p* < 0.05) between stress and non-stress conditions. The stress condition led to a significant reduction in both the conflict effect and conflict score, indicating that acute stress enhances the participants’ inhibitory control abilities. The result of the conflict effect and conflict score is shown in Figure 5.

### 3.3. EEG Results

#### 3.3.1. Frequency Domain Analysis

Paired *t*-tests were conducted to compare the power values of the delta, theta, and alpha frequency bands in the frontal region under stress and non-stress conditions. A significant difference was observed only in the beta band (t = −2.383, *p* < 0.05), with the power value of the beta band being significantly higher under stress conditions than under non-stress conditions. The topographic maps of the beta band power values under the two conditions are illustrated in Figure 6.

#### 3.3.2. Microstate Analysis

The topographic maps of the four dominant microstate classes under stress and non-stress conditions are depicted in Figure 6. These four classes accounted for 72.88% of the global variance in the stress group and 74.75% in the non-stress group. Paired-sample *t*-tests on the duration, occurrence, contribution, and transition probabilities between the four microstate classes under the two conditions revealed that, compared to the non-stress condition, the duration of microstate class D was significantly increased under stress (t = −3.322, *p* < 0.001). The duration (t = 3.082, *p* < 0.01), occurrence frequency (t = 2.760, *p* < 0.01), and contribution (t = 3.861, *p* < 0.001) of class C were significantly decreased under stress. Additionally, the transition probabilities from class B to D (t = −3.647, *p* < 0.01) and from class C to D (t = −2.905, *p* < 0.001) were increased under stress, while the transition probabilities from class B to C (t = 2.693, *p* < 0.05) and from class D to C (t = 2.857, *p* < 0.001) were decreased. The results of these indicators are shown in Figure 7.

### 3.4. Regression Analysis

A regression analysis was conducted to examine the relationship between the difference in frontal EEG beta band power values under non-stress and stress conditions and the differences in behavioral outcomes—precisely, the conflict effect and conflict score—between the two conditions. The results revealed a significant association (F = 4.389, *p* < 0.05) between the difference in frontal EEG beta band power values under stress and non-stress conditions and the conflict effect. The equation for the difference in the conflict effect between non-stress and stress conditions is (Difference in conflict effect) = 41.476 + 4.11 × (Difference in frontal EEG beta band power values between non-stress and stress conditions). This indicates that the more significant the impact of stress on the frontal beta band power, the more pronounced the enhancement of inhibitory control induced by stress.

## 4. Discussion

In this study, we induced acute stress in the laboratory using the SECPT paradigm to explore the impact of acute stress on individuals’ inhibitory control abilities. It further investigated the underlying mechanisms of this influence through frequency domain analysis and microstate analysis of resting state electroencephalogram (EEG) data. The findings of this study indicate that acute stress enhances an individual’s inhibitory control capabilities, a result attributed to the activation of the attentional network and the consequent elevation of attentional levels under acute stress conditions.

Initially, the increase in cortisol levels, heart rate, and negative emotions and the decrease in positive emotions following the SECPT, along with the subjective reports of stress experienced by the participants, demonstrate that the SECPT paradigm successfully induced acute stress in this experiment. Previous research has summarized various laboratory stress induction paradigms, with the SECPT being widely recognized for its ability to elicit physiological changes in participants [35]. Although the activation of the HPA axis may not be as pronounced as with the Trier Social Stress Test (TSST), and it is not as frequently employed like the TSST, the SECPT is advantageous due to its shorter duration and less stringent requirements for experimenters and experimental conditions. This study provides evidence supporting the SECPT as a viable laboratory paradigm for inducing acute stress.

Secondly, the majority of existing research indicates that acute stress can diminish inhibitory control [15,41]; however, a subset of studies has reported that stress might actually enhance this ability [16,19] or have no effect at all [42,43]. This study aligns with the perspective that acute stress can bolster inhibitory control, suggesting that individuals may fortify their inhibitory control mechanisms as a means of adapting to stressful environments. Despite a paucity of research on the neural mechanisms that facilitate stress’s enhancement of inhibitory control, the prevailing belief is that this phenomenon is contingent upon the level of stress experienced. While no definitive experiments have established a direct link between stress intensity and cognitive function, there is evidence to suggest that the relationship between stress-related hormone levels and cognitive performance is U-shaped in nature [44]. This implies that both minimal and extreme stress levels can be detrimental to cognition, whereas moderate stress levels may actually improve cognitive performance. Laboratory-induced stress is typically less intense than that encountered in real-life scenarios and is often within a range that individuals can manage. Consequently, it is posited that such manageable stress can facilitate enhanced inhibitory control performance. Beyond the stress intensity hypothesis, research has also indicated that stress might adjust the balance between speed and accuracy in task performance, stress could potentially elevate vigilance, prompting individuals to prioritize cautiousness over speed [45]. Although this study did not detect significant disparities in the accuracy rates of the Stroop task between the stressed and baseline conditions, this finding does not preclude the possibility that stress could influence the trade-off between speed and accuracy in performance.

This study elucidates the neurobiological underpinnings of the conclusions reached via frequency domain analysis and microstate analysis: it revealed significant increases in the power values of the beta band under stress, with beta waves typically associated with an individual’s level of attention. This suggests that the level of attention of the participants under stress conditions is higher compared to non-stress conditions. Additionally, microstate analysis indicated that stress significantly increased the duration of microstate class D and the transition probabilities from classes B and C to D. Class D is considered to be related to the negative BOLD activation in the suitable dorsal and ventral regions of the frontal and parietal lobes, which is significantly associated with the attention network [29]. This study provides an alternative perspective on the significant enhancement of the activation level of the attention network under stress and the substantial increase in the shift from the visual network (class B) and the salience network (class C) to the attention network under stress. Furthermore, regression analysis found that the increase in beta power values before and after stress was significantly correlated with the reduction in the conflict effect before and after stress. The more the beta power values increased, the more significant the reduction in the conflict effect, indicating a more significant enhancement of the individual’s inhibitory control ability under stress. The results suggest that changes in the activation of the attention network under stress are the reason for the impact of stress on the individual’s inhibitory control ability.

Previous research has explored the relationship between stress and attention, with mixed results. Some studies suggest that stress can impair attention [46], while others propose that stress can enhance an individual’s attention. This study provides evidence supporting the enhancement of attention under stress. The reasons for the varying results may be related to the type and intensity of the stressors, as well as individual differences such as gender, stress sensitivity, and psychological resilience: compared to stress in real environments, laboratory-induced stress is usually relatively weak, so low-intensity laboratory stress may enhance an individual’s level of attention, while high-intensity stress in real situations can impair attention; differences in stressor types mainly lie in the varying degrees of activation of the fast-acting sympathetic adrenal medulla (SAM) axis and the slower hypothalamic pituitary adrenal (HPA) axis: the SAM axis primarily leads to the release of catecholamines, causing changes in heart rate and blood pressure, while the HPA axis mainly results in cortisol release, a process that is slower overall. Some studies suggest that pure cold pressor stress can produce heart rate responses without cortisol responses [47]. In contrast, sociopsychological stressors, such as the TSST and SECPT, which involve the social evaluation of participants, tend to produce more robust activation of both the SAM and HPA axes [48]. Some research indicates that, under cold pressor stress, the SAM axis is activated. Still, there is no significant cortisol response, meaning the HPA axis is not significantly activated, leading to clear improvements in attention and memory [47]; other studies using the TSST as a stress induction paradigm find significant activation of both the SAM and HPA axes but no improvement in individual attention and learning [49]. Therefore, different stressor types and varying degrees of activation of the two axes may lead to changes in hormone levels, which, in turn, affect different brain networks. Additionally, the impact of stress on individuals may vary: some studies suggest that stress can regulate the locus coeruleus-norepinephrine (LC-NE) system, which is influenced by estrogen; thus, women’s arousal levels may be more susceptible to stress than men’s [50]. Many factors affect the relationship between stress and attention, and future studies could vary the stressor or participant characteristics to see if the results change.

This study also uncovered an intriguing phenomenon: although individuals demonstrated improved inhibitory control performance under stress conditions, microstate analysis revealed that the duration, frequency, and coverage of microstate class C were significantly lower under stress compared to non-stress conditions. Class C is not only associated with the activation of the salience network (SN), which involves the dorsal anterior cingulate cortex, bilateral inferior frontal gyrus, and insula, but is also related to the transition between the central executive network and the default mode network (DMN) [29,51]. The SN plays a crucial role in cognitive events, guiding individual behavior, and is also involved in the detection and integration of emotional and sensory stimuli [52]. The default network, dorsal attention network, and salience network are three core brain networks that have garnered widespread attention in research; they are essential for maintaining normal psychological states and cognitive abilities (such as attention, working memory, decision-making, etc.) throughout an individual’s development and aging process [53]. Therefore, any suppression or damage to the activation of these three brain networks could impact cognitive function, and theoretically, an individual’s inhibitory control ability would be compromised [54]. Thus, the microstate results seem theoretically contradictory to the behavioral findings. We have discussed why this might be the case, suggesting that, while stress in the experiment indeed significantly reduced the individual’s salience and inhibitory network activation, thereby diminishing their inhibitory capacity, the enhancement of the attention network activation due to stress outweighed the negative effects of the decreased activation in the inhibitory network. Consequently, this was manifested behaviorally as an increased conflict effect in the Stroop task and an enhancement of the inhibitory control ability. If the stressor or participant conditions were altered, it might tip the scales, leading to a decrease in the individual’s behavioral inhibitory control performance due to the greater impact of reduced salience network activation over the increased activation of the attention network. Future studies could vary the intensity of the stressor to test this hypothesis.

This study utilized a within-subjects design, which inherently carries the risk of practice effects. To counteract this potential bias, it was essential to take measures to minimize these effects. While many stress-related studies within this design framework opt for a 24-h interval to mitigate the practice effects [55,56], this study opted for a more rigorous approach. Specifically, a minimum of one week was scheduled between the two experimental sessions to effectively eliminate any carryover effects and ensure the validity of the findings [57,58]. Additionally, to account for potential order effects inherent in within-subjects designs, participants were subjected to two conditions of experimental order randomly. Since most of the participants were not from our institution and had not previously participated in psychological experiments, they were not familiar with the procedures. To prevent differences in task performance due to variations in experimental content, we instructed participants in both conditions to respond as quickly and accurately as possible during the Stroop task.

The study also has some limitations:

Firstly, the analysis was conducted on a final sample size of 32 individuals, which is modest. To lend greater credence to our findings, future studies should aim to expand this sample size.

Secondly, the study’s participants were carefully selected for homogeneity, all being male college students from various universities with comparable ages and educational backgrounds. The exclusion of females was a deliberate choice to control for the confounding effects of physiological cycles, which can cause significant fluctuations in estrogen hormone levels. Research has established that these hormonal levels are intricately linked to the activity of the hypothalamic–pituitary–adrenal (HPA) axis [59,60] and are correlated with stress-related symptomatology [61]. Moreover, there is evidence that sex hormones from the hypothalamus–pituitary–gonadal (HPG) axis and glucocorticoids from the HPA axis interact, exerting a combined influence on individual behavior [62]. Furthermore, prior research has shown that the effects of stress on inhibitory control differ significantly between male and female subjects [17]. Consequently, this study deliberately excluded female participants to avoid potential confounds. Future research endeavors should aim for a more diverse participant pool, spanning a wider range of ages and educational levels. It would also be advantageous to include postmenopausal women or women in phases outside of their follicular, premenstrual, and menstrual cycles to enrich the study’s demographic scope.

In addition, the study only employed the Stroop task for research. It would verify the findings if the experiment used a variety of other inhibitory control tasks.

Lastly, the stress paradigm used in the study is a commonly employed laboratory-induced stress model. It is widely acknowledged that laboratory-induced stress is generally mild to moderate and does not match the intensity of stress encountered in real-life situations. Therefore, future research could utilize real-life scenarios or advanced VR technology to induce more intense stress for further investigation.

## 5. Conclusions

Inhibitory control stands as a pivotal aspect of executive function, a faculty that stress, a constant in our daily lives and professional arenas, can influence. Our study, which involved the induction of acute stress, has uncovered a capacity for stress to bolster an individual’s inhibitory control capabilities. A deeper dive into the resting state EEG has shed light on the mechanism behind this phenomenon: stress appears to amplify the activation of the attention network. There is a direct correlation between the stress-induced rise in an individual’s attention-related beta power and the enhancement of their inhibitory control performance. Moreover, our study, leveraging microstate analysis, has observed an extension in the duration of category D among individuals subjected to stress. This finding further supports the notion of an elevated activation of the attention network under stress. The existing body of research on stress-related microstates is not abundant, and our work aims to fill this gap. By examining the impact of stress on inhibitory control through the lens of microstate analysis, we have shed new light on how stress might augment inhibitory control. Our analysis suggests that the beneficial effect of stress on inhibitory control is not mediated through the executive function-related network but rather by increasing the individual’s attentiveness. This insight paves the way for novel research directions in stress-related research.

## Figures and Tables

**Figure 1 brainsci-14-01013-f001:**
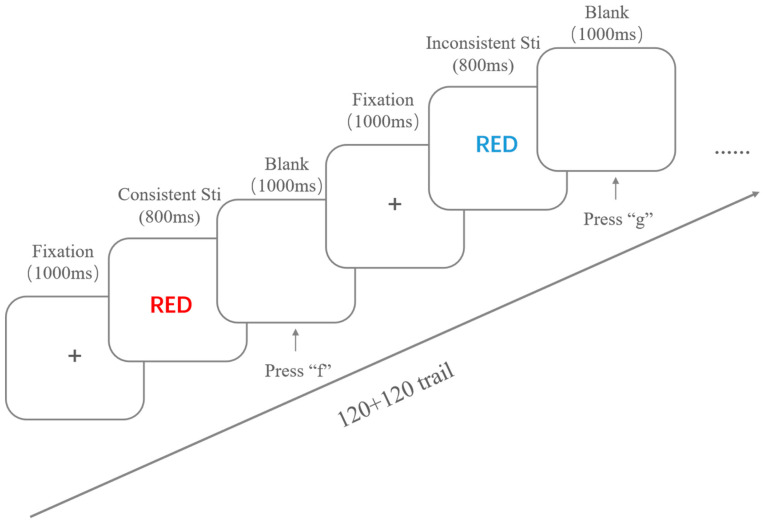
The specific procedure of the Stroop task. The term "RED" in red color represents a consistent stimulus, while "RED" in blue color denotes an inconsistent stimulus.

**Figure 2 brainsci-14-01013-f002:**
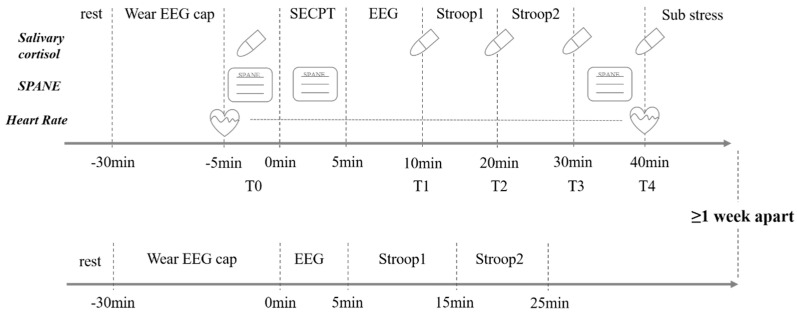
Experimental procedure (The diagram illustrates the timing of heart rate measurements, cortisol assessments, and the administration of the SPANE scale in graphical form, as well as the specific procedures for both the stress and non-stress experiments.).

**Figure 3 brainsci-14-01013-f003:**
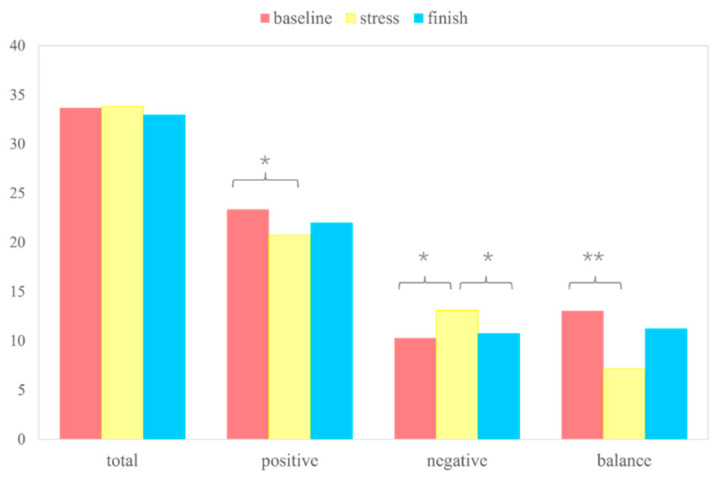
SPANE score, total, positive, negative, and balanced score at the time of pre-stress, post-stress, and post-experiment (*: *p* < 0.05, **: *p* < 0.01).

**Figure 4 brainsci-14-01013-f004:**
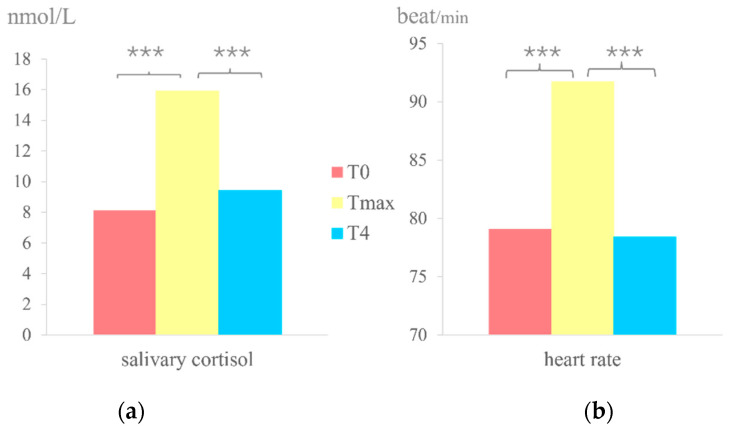
(**a**) The result of salivary cortisol under pre-stress, peak, and post-stress conditions (***: *p <* 0.01). (**b**) The result of heart rate in pre-stress, peak, and post-stress conditions (***: *p <* 0.01).

**Figure 5 brainsci-14-01013-f005:**
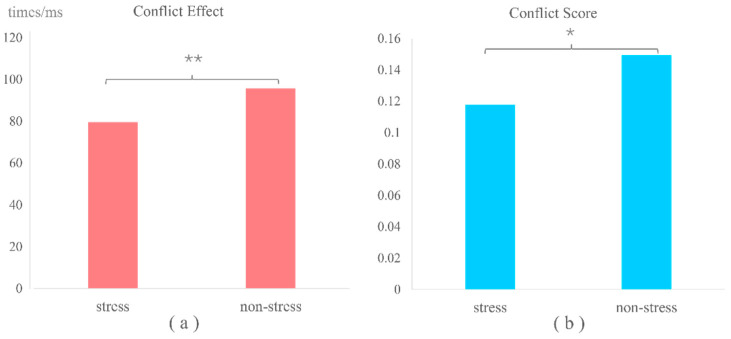
The results of the conflict effect and conflict score under stress and non-stress conditions. (**a**) The differences in conflict effect in stress and non-stress conditions (**: *p <* 0.01), and (**b**) the differences in conflict score in stress and non-stress conditions (*: *p <* 0.05).

**Figure 6 brainsci-14-01013-f006:**
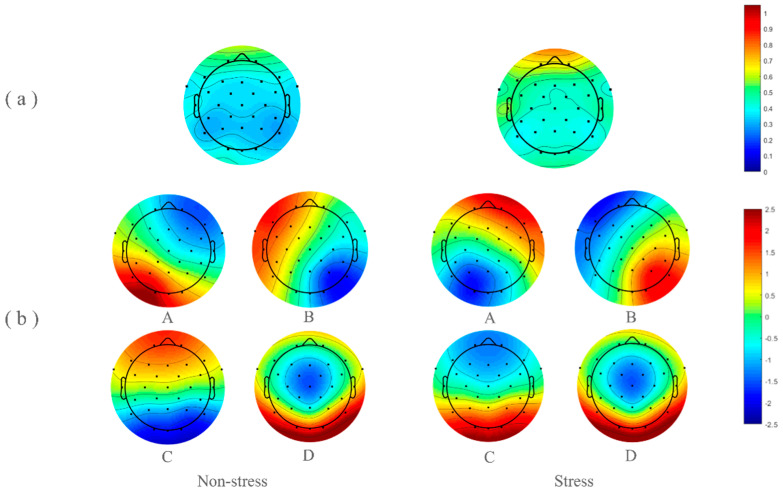
The topographic maps of the beta band power and the topographic maps of the four dominant microstate classes under stress and non-stress conditions. (**a**) The topographic maps reveal significant differences in the power of the beta band in the frontal region. (**b**) A–D represent the four categories of microstate status respectively. The four maps on the left show the topographic maps of the four dominant microstate classes under non-stress conditions; the four maps on the right show the topographic maps of the four dominant microstate classes under stress conditions.

**Figure 7 brainsci-14-01013-f007:**
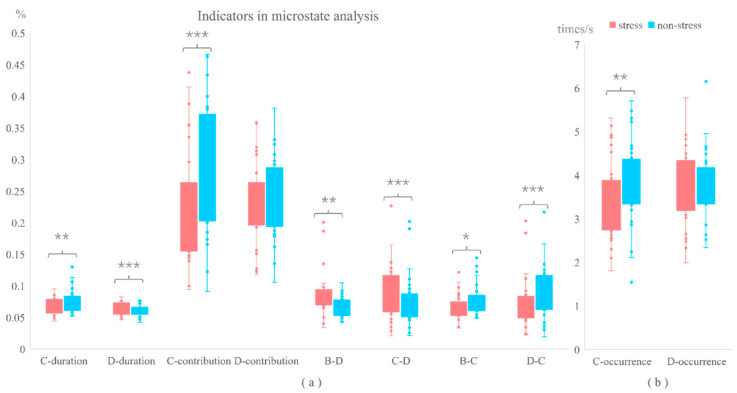
The result of indicators in a microstate. (**a**) Duration and contribution of class C show differences in stress and non-stress conditions (**: *p* < 0.01; ***: *p* < 0.001); the duration of class D shows differences in the stress and non-stress conditions (***: *p* < 0.001); the transition probabilities of B–D, C–D, B–C, and D–C show differences in stress and non-stress conditions (*: *p* < 0.05; **: *p* < 0.01; ***: *p* < 0.001). (**b**) Occurrence of class C show differences in the stress and non-stress conditions (**: *p* < 0.01).

## Data Availability

The data presented in this study are available on request from the corresponding author. The reason why the data cannot be published is that the experiment has not been fully completed yet, and the data are temporarily not suitable for public disclosure.

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
