# Peer review of "EEG Evidence of Acute Stress Enhancing Inhibition Control by Increasing Attention"

_brainsci, 2024, doi:10.3390/brainsci14101013_

Round 1

Reviewer 1 Report

Comments and Suggestions for Authors

This is an interesting study investigating the effects of acute stress on inhibitory control and attentional processes using EEG data. The paper is well-written and I agree that it may contribute to the literature well. However, there are some room for improvement:

1. The theoretical foundation for why stress would enhance rather than impair inhibitory control is not sufficiently developed. The authors should provide a stronger theoretical rationale and literature review, especially considering most research finds that stress impairs cognitive performance.

2. The study uses a sample of only 32 male college students. The small sample size should be acknowledged and highlighted.

3. The justification for excluding female participants due to hormonal fluctuations should be elaborated upon, and the paper should discuss how future studies might address this gap.

4. I like the use of within-subject design. However, the within-subject design, while very useful, is vulnerable to practice effects. Although the authors mention a one-week gap between sessions, this is debatable. The paper should explain more thoroughly how practice effects were controlled. I would suggest the author to cite a few studies that use similar within-subject design with 1 week of washout period  and show that it is a norm and spillover from practice effect is less likely. One quick search and I found a few papers on within-subject design that are relevant: The effect of state gratitude on cognitive flexibility: A within-subject experimental approach. Brain Sciences, 10(7), 413 and Brief mindfulness breathing exercises and working memory capacity: Findings from two experimental approaches. Brain Sciences, 11(2), 175.

5. The microstate analysis results are interesting but somewhat contradictory to the behavioral findings. Specifically, why would stress reduce the activation of the salience network (class C) while enhancing attention and inhibitory control? The authors acknowledge this contradiction but do not provide a clear explanation. A more detailed discussion is needed on this discrepancy

6. Furthermore, the transition probabilities between microstates are reported, but the functional implications of these transitions remain unclear.

7. The description of EEG data preprocessing and artifact removal is adequate but lacks specific details about how noise was handled.

Author Response

Thank you for your suggestion, sorry for taking up your time, here is my answer and my revised article in appendix.

Comments 1: The theoretical foundation for why stress would enhance rather than impair inhibitory control is not sufficiently developed. The authors should provide a stronger theoretical rationale and literature review, especially considering most research finds that stress impairs cognitive performance.

Response 1: I've included the potential reasons for the stress-induced enhancement of inhibitory control in the discussion section; for details, see page 11, line 356-371. Additionally, the EEG analysis performed in this study was designed to investigate the mechanisms underlying this finding. Consequently, the EEG results can be considered as part of the explanation for how stress might enhance inhibitory control.

Comments 2: The study uses a sample of only 32 male college students. The small sample size should beacknowledged and highlighted.

Response 2: The limitation of small sample size is added in Disgussion, seen in page 13, line 438.

Comments 3: The justification for excluding female participants due to hormonal fluctuations should be elaborated upon, and the paper should discuss how future studies might address this gap.

Response 3: Due to the influence of the menstrual cycle on women, estrogen levels fluctuate significantly at different times. In addition, sex hormone levels are regulated by stress and can affect an individual's cognitive activities. These aspects have been thoroughly described in the discussion section; for details, please refer to page 13, line 440-451.

Comments 4:  I like the use of within-subject design. However, the within-subject design, while very useful, is vulnerable to practice effects. Although the authors mention a one-week gap between sessions, this is debatable. The paper should explain more thoroughly how practice effects were controlled. I would suggest the author to cite a few studies that use similar within-subject design with 1 week of washout period  and show that it is a norm and spillover from practice effect is less likely. One quick search and I found a few papers on within-subject design that are relevant: The effect of state gratitude on cognitive flexibility: A within-subject experimental approach. Brain Sciences, 10(7), 413 and Brief mindfulness breathing exercises and working memory capacity: Findings from two experimental approaches. Brain Sciences, 11(2), 175.

Response 4: In the realm of stress and cognitive function research, certain within-subjects studies have measured cognitive performance solely before and after stress exposure, neglecting the potential for carry-over effects. Although some literature posits that a 24-hour interval might be adequate to dissipate such effects, a more rigorous standard advocates for a separation of at least one week between experimental sessions. In line with this stringent criterion, our study has chosen to implement a more rigorous approach to minimize carry-over effects. For an exhaustive account and pertinent citations, please refer to page 13, line 427-436.

Comments 5: The microstate analysis results are interesting but somewhat contradictory to the behavioral findings. Specifically, why would stress reduce the activation of the salience network (class C) while enhancing attention and inhibitory control? The authors acknowledge this contradiction but do not provide a clear explanation. A more detailed discussion is needed on this discrepancy.

Response 5: It is widely accepted that microstate category C is associated with the executive function network, while category D is related to the attention-related network. The study's findings indicate an enhancement in the relevant metrics of category D, further corroborating the results of frequency domain analysis: stress leads to increased activation of the attention-related network, thereby promoting inhibitory control performance. Conversely, the metrics related to category C have decreased, suggesting that under stress, the activation of networks associated with executive functions is reduced. I interpret this as follows: stress indeed causes a reduction in the activation of networks related to inhibitory control, but due to the increased activation of the attention-related network, and given that in this experiment, the positive impact of the attention-related network activation on inhibitory control behavior outweighs the negative impact of the deactivation of the inhibitory control network, the behavioral manifestation is a facilitative effect. As the intensity of stress changes, this relationship may shift, potentially revealing an inhibitory effect of stress on behavioral performance. This content has been elaborated upon in the discussion section, seen in page 13, line 418-426.

Comments 6: Furthermore, the transition probabilities between microstates are reported, but the functional implications of these transitions remain unclear.

Response 6: Research on microstates is still relatively scarce, with the majority focusing on the duration, occurrence, and contribution of various categories. Transition probability, which refers to the likelihood of a microstate transitioning to another, provides an alternative perspective on the increase or decrease in the occurrence of the four microstate categories. An increase in the transition probability from categories B to D, and a decrease in the transition probability from C to other categories, indicate an increased proportion of category D. Conversely, a decrease in the transition probability from categories B and D to C, coupled with an increase in the transition probability from C to other categories, suggests a reduced proportion of category C.

Comments 7: The description of EEG data preprocessing and artifact removal is adequate but lacks specific details about how noise was handled.

Response 7: Typically, EEG noise generates excessive frequencies and floating amplitudes; therefore, noise removal involves eliminating segments with large floating amplitudes and high frequencies. Details have been added on page 6, line 207-209 of the article.

Reviewer 2 Report

Comments and Suggestions for Authors

EEG Evidence of Acute Stress Enhancing Inhibition Control by Increasing Attention

Introduction: The authors introduced acute stress and HPA axis activation. This sentence needs to be deeper explained since it is quite vague. Similarly, the statement “Psychologically, stress affects the individual's brain, leading to various cognitive and emotional changes (Shansky & Lipps, 2013; Shields et al., 2024).” It is used to introduce inhibitory control (that is an executive function). However, this link needs to be reformulated in a better way. Acute stress could affect cognitive (executive, cold functions), but also emotions (hot functions). Moreover, the symptoms (obesity, craving, etc.) may refer to a chronic stress condition. Please, revise these statements, clarify the symptoms and alterations linked to acute stress, and add the related references.

The authors need to clarify the use of the terms (technology, EEG technique, etc.) in the paragraph about EEG microstates. The authors need to be more specific about the fact that is an analytical, sophisticated technique that allows to analyze EEG signals and captivates information that is not possible with our techniques. Moreover, the distinction between fMRI and EEG needs to be improved. Therefore, I suggest adding a definition of microstates, according to psychophysiology.

Moreover, and I have appreciated this, the authors collected and determined the salivary cortisol levels. A brief description of previously published evidence about cortisol levels and acute stress/ executive functions is also needed. Please, add.

The aims at the end of the section are quite vague. Please, reformulate them in light of the existing studies and the theoretical motivations added in the introduction.

 The methods are well-described and the figures are also informative, about the procedure and the setting. In the data analysis , please clarify “The scores for subjective stress levels were analyzed using a one-sample t-test.”

The results, despite their complexity, are interesting. The authors performed a regression, but a correlation between heart rate and cortisol levels should be interesting.

 Discussion: Please clarify and add references “Some studies further differentiate inhibitory control into response inhibition and conflict inhibition, proposing that stress does not affect response inhibition but does reduce the ability to resolve conflicts. This study presents findings that contradict the behavioral outcomes of most previous research” In this statement you need to be more specific about “resolve conflicts”, and add more references about. Similarly the link between the obtained results and the DAN should be improved. Moreover, the role played by DMN needs to be revised and more references are needed. Indeed, DMN role was revised during the last years; it is not only task-negative but plays also an active role, for cognitive /emotional functions. Moreover, the link between SN and DMN needs to be revised.

Please, clarify this “Due to the influence of hormone levels, female participants were not included in the study.” Quite vague. However, I know that you mean the hormonal levels in the normally cycling women and in those who use hormonal contraceptives. 

Author Response

Thank you for your suggestion, sorry for taking up your time, here is my answer and my revised article in appendix.

Comments 1: Introduction: The authors introduced acute stress and HPA axis activation. This sentence needs to be deeper explained since it is quite vague. Similarly, the statement “Psychologically, stress affects the individual's brain, leading to various cognitive and emotional changes (Shansky & Lipps, 2013; Shields et al., 2024).” It is used to introduce inhibitory control (that is an executive function). However, this link needs to be reformulated in a better way. Acute stress could affect cognitive (executive, cold functions), but also emotions (hot functions). Moreover, the symptoms (obesity, craving, etc.) may refer to a chronic stress condition. Please, revise these statements, clarify the symptoms and alterations linked to acute stress, and add the related references.

Response 1: I have revised the introduction, adding descriptions of the physiological responses to stress and references related to its impact on cognitive functions. For details, please refer to the first paragraph of the introduction section(line 31-47).

Comments 2: The authors need to clarify the use of the terms (technology, EEG technique, etc.) in the paragraph about EEG microstates. The authors need to be more specific about the fact that is an analytical, sophisticated technique that allows to analyze EEG signals and captivates information that is not possible with our techniques. Moreover, the distinction between fMRI and EEG needs to be improved. Therefore, I suggest adding a definition of microstates, according to psychophysiology.

Response 2: Thanks for your suggestion. I have included the relevant definitions of microstates and corresponding references in the introduction section. Given the complexity and cost associated with fMRI, we intend to investigate changes in brain networks related to certain regions under stress conditions by examining the topographical terrain of microstates. For specific details on the addition of microstate explanations, please refer to page 2-3, line 75-94.

Comments 3: Moreover, and I have appreciated this, the authors collected and determined the salivary cortisol levels. A brief description of previously published evidence about cortisol levels and acute stress/ executive functions is also needed. Please, add.

Response 3: Because cortisol is a relatively easy-to-measure indicator of HPA axis activation levels, many stress studies have used cortisol response to verify whether stress has been induced, such as“Acute stress influences the emotional foundations of executive control: Distinct effects on control-related affective and cognitive processes”, “The discrepant effect of acute stress on cognitive inhibition and response inhibition”, “Acute stress impacts reaction times in older but not in young adults in a fanker task” and so on.

Comments 4: The aims at the end of the section are quite vague. Please, reformulate them in light of the existing studies and the theoretical motivations added in the introduction.

Response 4: Thank you for your feedback. I have outlined the majority of the future experimental research objectives in the discussion section, while the conclusion section now focuses solely on summarizing the results of this study to facilitate easier comprehension for readers. I have made the necessary revisions to the conclusion section, which can be found there.

Comments 5: The methods are well-described and the figures are also informative, about the procedure and the setting. In the data analysis , please clarify “The scores for subjective stress levels were analyzed using a one-sample t-test.”

Response 5: Due to the final paragraph of section 2.2.2 and the last sentence of the "Acute Stress Experiment" section in 2.3 both explaining that after the stress experiment, participants were asked whether they felt nervous during the stress induction process and to rate the degree of nervousness and pressure on a scale from 1 to 10, a one-sample t-test was conducted to see if the participants subjectively felt stressed, serving as a measure of stress induction.

Comments 6: The results, despite their complexity, are interesting. The authors performed a regression, but a correlation between heart rate and cortisol levels should be interesting.

Response 6: Currently, no correlation has been found between heart rate responses and cortisol responses. It is understandable that there are mechanisms differences between the SNS axis and the HPA axis, which may make it difficult to find significant correlations between the physiological indicators of their activation.

Comments 7:  Discussion: Please clarify and add references “Some studies further differentiate inhibitory control into response inhibition and conflict inhibition, proposing that stress does not affect response inhibition but does reduce the ability to resolve conflicts. This study presents findings that contradict the behavioral outcomes of most previous research” In this statement you need to be more specific about “resolve conflicts”, and add more references about. Similarly the link between the obtained results and the DAN should be improved. Moreover, the role played by DMN needs to be revised and more references are needed. Indeed, DMN role was revised during the last years; it is not only task-negative but plays also an active role, for cognitive /emotional functions. Moreover, the link between SN and DMN needs to be revised.

Response 7:  Thank you for your suggestion, here is my answer.

â‘  I have removed the section on the differences in inhibitory control task classifications that may account for discrepancies in research findings. Although there is research that has analyzed this aspect (titled "The discrepant effect of acute stress on cognitive inhibition and response inhibition"), the primary reasons here may have more to do with the stressor and the individual's speed-accuracy trade-off. This has been revised in the article; for details, please refer to page 11-12, line 356-371.

â‘¡ Thank you for your prompt. Regarding the definition of the Default Mode Network (DMN) and the associated references, I have included them in the article; for details, please see page X, line XX. Furthermore, I have added a discussion on the relationship between the DMN, the Dorsal Attention Network, and the Salience Network; you can find this on page 13, line 407-426.

Comments 8: Please clarify this “Due to the influence of hormone levels, female participants were not includedin the study." Quite vague. However, l know that you mean the hormonal levels in the normallycycling women and in those who use hormonal contraceptives.

Response 8: I have added a discussion in the 'Limitations' section of the research regarding the large fluctuations in hormone levels due to the menstrual cycle in women, which make them unsuitable for inclusion in experiments where hormone levels are used as an indicator to measure whether stress has been induced. For specifics, please see page 13, line 440-451.

Round 2

Reviewer 1 Report

Comments and Suggestions for Authors

The authors have sufficiently addressed all my comments.

Reviewer 2 Report

Comments and Suggestions for Authors

The authors improved the manuscript.